# CCUS Technology and Carbon Emissions: Evidence from the United States

Min Thura Mon [1], Roengchai Tansuchat [1,2,*] and Woraphon Yamaka [1,2]

1   Faculty of Economics, Chiang Mai University, Chiang Mai 50200, Thailand;
    minthuramon_m@cmu.ac.th (M.T.M.); woraphon.yamaka@cmu.ac.th (W.Y.)
2   Centre of Excellence in Econometrics, Faculty of Economics, Chiang Mai University,
    Chiang Mai 50200, Thailand
*   Correspondence: roengchai.tan@cmu.ac.th

**Abstract:** Carbon Capture, Utilization, and Storage (CCUS) represents a vital technology for addressing pressing global challenges such as climate change and carbon emissions. This research aims to explore the relationship between the CCUS capability and carbon emissions in the United States considering thirteen predictors of CCUS and carbon emissions. Incorporating these predictors, we aim to offer policymakers insights to enhance CCUS capabilities and reduce carbon emissions. We utilize diverse econometric techniques: OLS, Lasso, Ridge, Elastic Net, Generalized Method of Moments, and Seemingly Unrelated Regression. Elastic Net outperforms the other models in explaining CCUS, while OLS is effective for carbon emissions. We observe positive impacts of the number of projects and foreign direct investment on the CCUS capacity, but limited influence from the CCUS technology level. However, the relationship between the CCUS capacity and carbon emissions remains limited. Our study highlights the importance of incentivizing projects to increase CCUS capabilities and recognizes the critical role of legal and regulatory frameworks in facilitating effective CCUS implementation in the US. Moreover, we emphasize that achieving decarbonization goals necessitates the development of affordable green alternatives. It is essential to view CCUS as a complementary, rather than a sole, solution for emission reduction as we work towards achieving net-zero emission targets.

**Keywords:** CCUS; CCUS capacity; carbon emissions; macroeconomy; energy consumption

## 1. Introduction

Despite ongoing efforts to address climate change, the global consumption of fossil fuels, including coal, oil, and natural gas, continues to rise, and consequently, the emissions of greenhouse gases, primarily carbon dioxide ($CO_2$), remain alarmingly high [1]. Across the globe, countries have established ambitious objectives for carbon neutrality and net-zero emissions aimed at mitigating the effects of climate change. In the endeavor to reach net-zero objectives, one highly effective approach involves the deployment of Carbon Capture, Utilization, and Storage technologies, commonly referred to as CCUS. CCUS represents a comprehensive suite of technologies that holds great potential for making substantial contributions to global energy and climate targets in diverse ways. It is widely recognized as a pivotal clean technology, a viewpoint shared by experts, including officials at IEA [2].

The CCUS capacity refers to the collective capability of CCUS technologies and projects to capture and store $CO_2$ emissions. It includes the total capacity for active and commercial projects across various CCUS facilities while excluding suspended and decommissioned projects as well as the transport capacity of each project [3]. This capacity is measured in terms of the amount of $CO_2$ that can be effectively captured and stored, typically expressed in metric tons (or million metric tons) of $CO_2$ per year. The CCUS capacity is a crucial metric

in efforts to reduce $CO_2$ emissions and combat climate change, as it represents the potential for mitigating greenhouse gas emissions from industrial processes and power generation.

Within the realm of CCUS, significant sources of $CO_2$ emissions originate from power generation facilities, whether they are fueled by biomass or fossil resources. Furthermore, CCUS encompasses the potential for the direct capture of $CO_2$ from the atmosphere. Once captured, the compressed $CO_2$ can be transported through a variety of channels, including pipelines, ships, railways, or trucks, and subsequently employed for various applications if it is not immediately used upon capture. Additionally, CCUS offers the option of injecting $CO_2$ into deep geological formations, such as saline reservoirs or depleted oil and gas reservoirs, providing a secure and enduring repository for the stored $CO_2$ [4]. Figure 1 provides a schematic representation of the CCUS value chain. Currently, CCUS facilities operating worldwide possess the capacity to abate more than 40 million metric tons (Mt) of $CO_2$ equivalent annually.

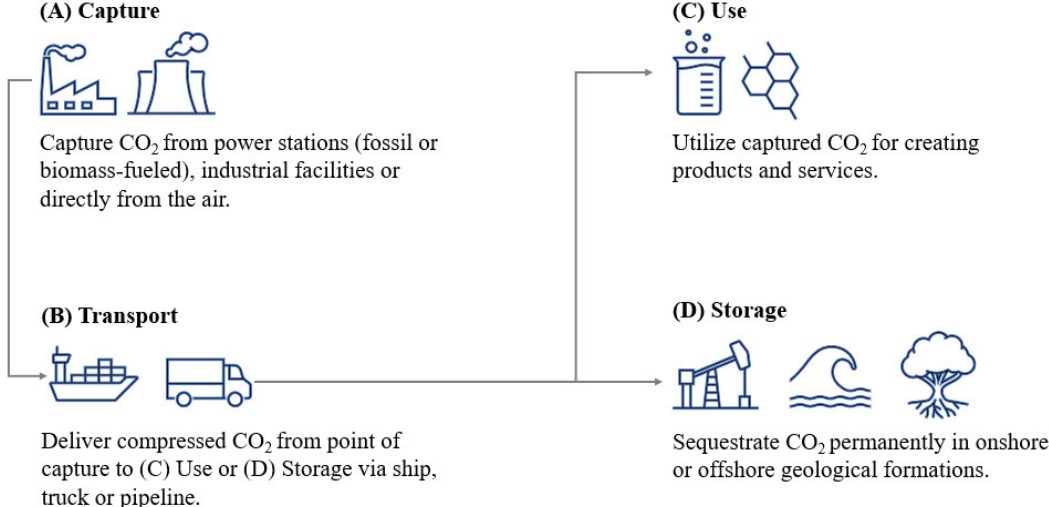

**Figure 1.** Schematic view of the full CCUS value chain.

While CCUS project deployment is becoming increasingly diverse across various regions, a significant concentration of global CCUS projects is evident in the United States, as depicted in Figure 2. Among nations, the United States has assumed a leadership position in CCUS, with a full-chain capacity of 21.89 Mt of $CO_2$ and 16 active projects as of 2022. This accounts for approximately 50.9% of the global capacity in that year [5].

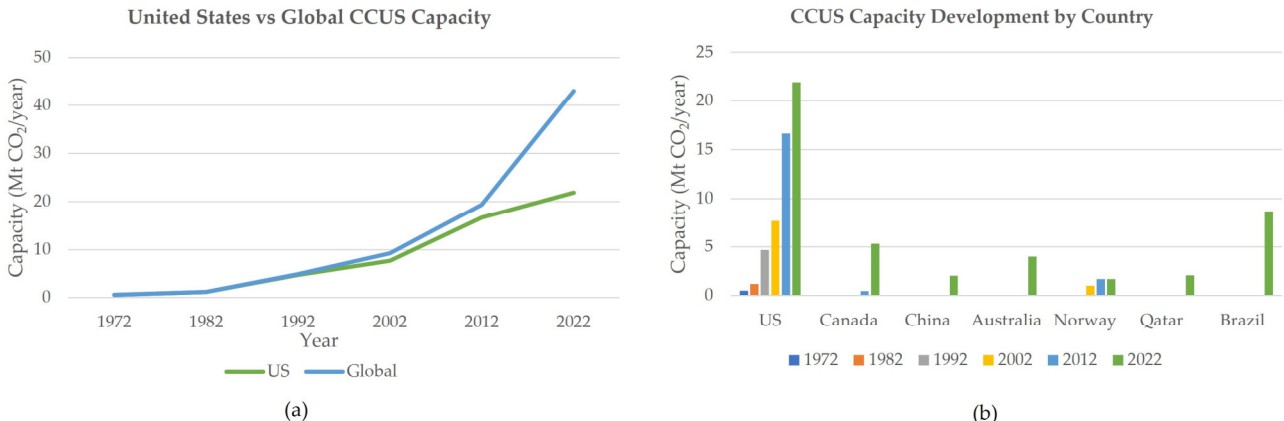

**Figure 2.** (**a**) Capacity comparison of CCUS facilities in the US and globally over 50 years and (**b**) CCUS projects in terms of its capability by each country [3].

While historical disparities between CCUS deployment and expectations have been notable, there has been a significant surge in momentum in recent years, as evident in Figure 2. Across the entire global CCUS value chain, there are currently more than 500 projects in various stages of development. As of early 2022, CCUS project developers have set ambitious targets for approximately 50 new CCUS facilities to become operational by 2030. If this target is met, it could result in the annual capture of approximately 125 Mt of $CO_2$. However, even with this level of progress, the overall capacity of CCUS would still fall significantly short of the one-third of 1.2 Bt of $CO_2$ per annum that the 2050 Scenario of Net Zero Emissions calls for [5].

Despite the recognized potential of CCUS technologies in reducing greenhouse gas emissions, their widespread implementation remains limited. Several factors contribute to the limited implementation of CCUS projects. This includes high costs associated with CCUS technologies acting as a deterrent, compounded by insufficient financial incentives [6] and supportive government policies [7], complex regulatory requirements, technical challenges, and public perception concerns further hindering adoption [7,8]. Additionally, the availability of alternative emission reduction methods and limited awareness about CCUS benefits contribute to the issue [9].

From the above, the relationship between CCUS adoption and emissions remains inadequately understood, leaving a gap in our comprehension of its effectiveness. While the engineering, scientific, and technological aspects of CCUS have been extensively explored [2], they may not offer a comprehensive view, particularly from an economic standpoint. This study emphasizes the urgent need to delve deeper into the barriers hindering CCUS deployment and how macroeconomic factors, energy consumption, and energy prices influence the feasibility of CCUS projects and their impact on carbon emissions in the United States—a nation recognized for its leadership in certain CCUS technology and implementation domains. Moreover, despite existing research focusing on the techno-economic analysis of CCUS projects [10–12] and general equilibrium models [13], empirical econometric research on CCUS capacity remains notably lacking [14]. Therefore, this study employs various econometric modeling techniques, including Ordinary Least Squares (OLS), Lasso, Ridge, Elastic Net, Generalized Method of Moments, and Seemingly Unrelated Regression (SUR), to uncover crucial insights into the link between CCUS capacity and carbon emissions and their associated factors, aiming to strengthen the robustness of the results and contribute significantly to the existing body of knowledge.

This paper's structure consists of five sections. Section 1 introduces the research topic, while Section 2 offers a comprehensive literature review encompassing CCUS, carbon emissions, energy consumption, economic activity, and their interconnectedness. In Section 3, the methodology is outlined, encompassing the use of econometric methods, data description, and empirical model. Moving to Section 4, this part of the paper presents and discusses the analytical results. Lastly, Section 5 encapsulates this study's conclusion, summarizing key findings and implications.

## 2. Literature Review

Numerous scholars have diligently identified various factors influencing carbon emissions, such as finance, policy, and urbanization [15,16]. However, many studies have overlooked the impact of CCUS, including its capacity, technology, and development [17–20]. Few studies, especially quantitative ones, have explored the relationship between CCUS technology and carbon emissions [14,21] and the full-chain version of CCUS is more common than individual segments such as capture, storage, and utilization, as in Table 1.

Alsarhan et al. [22] stated that CCUS technology can remove carbon dioxide from the atmosphere, essentially acting as a "carbon sink". This process would result in a reduction in carbon emissions present in the air. However, some researchers have asserted that the success of CCUS in reducing carbon emissions may be hindered by several factors. These include high failure rates, a lack of financial support and market incentives, and an inadequate regulatory framework [22–24]. After accounting for technological regularity

and economic benefits, Wang et al. [25] concluded that uncertainty remains regarding CCUS technology's ability to effectively reduce carbon emissions. Furthermore, the high cost of utilizing CCUS technology may hinder the emission reduction of $CO_2$ [26]. It is essential to recognize that the costs, benefits, and overall maturity of CCUS technology can vary significantly at different stages of development because it is developed following the technological phases [27]. As a result, the effect of CCUS technology on reducing emissions may change depending on the stage of development. For instance, the initial stages of CCUS innovation may have a more detrimental effect on carbon emission efficiency than in later stages due to elevated costs and energy consumption for technological implementation. Zhang et al. [14] discovered that China's digital economy serves as a favorable moderator, enhancing the impact of increased carbon emission efficiency through CCUS technology.

**Table 1.** Global CCUS capacity across its value chain [3].

| Year | Capture (Mt $CO_2$/Year) | Utilization (Mt $CO_2$/Year) | Storage (Mt $CO_2$/Year) | Full Chain (Mt $CO_2$/Year) |
|---|---|---|---|---|
| 2018 | 0 | 0.838 | 0 | 36.8769 |
| 2019 | 0.3 | 0.838 | 0 | 42.9769 |
| 2020 | 1.78 | 0.838 | 1.12 | 42.8769 |
| 2021 | 1.78 | 0.838 | 1.12 | 41.8309 |
| 2022 | 1.78 | 0.963 | 1.12 | 43.0109 |

Note: Transport capacity is included under full chain. The data cover CCUS projects under operation status.

For instance, a higher GDP per capita may stimulate investment in CCUS projects and research [23]. Similarly, improved employment rates resulting from CCUS initiatives can spur investment in CCUS projects [28]. Additionally, factors such as trade volume and foreign direct investment can influence a nation's industrial activity and emissions, thereby shaping the prioritization of carbon reduction technologies, including CCUS [29]. Moreover, the relationship between energy factors, such as consumption and prices, and CCUS adoption is complex. Energy consumption, particularly from fossil fuels, directly contributes to carbon emissions [30], making it a critical consideration for CCUS implementation. Higher fossil fuel prices may provide incentives for CCUS investment, while lower prices could pose obstacles to adoption [31]. Furthermore, the consumption of renewable energy may impact the urgency of CCUS implementation, particularly for industries reliant on fossil fuels [32].

Considering the factors related to carbon emissions, researchers have dedicated their efforts to uncovering the relationship between macroeconomics, energy consumption, climate change, and emissions [33–35]. However, depending on the country and region, the impact of economic activity and energy consumption on carbon emissions can yield inconsistent effects and conflicting outcomes [36]. Some studies suggest that as a nation undergoes economic development and experiences a rise in energy consumption, its emissions tend to increase [37,38]. Conversely, other research indicates that increased economic activity and energy usage can lead to emission reductions. For instance, Sterpu et al. [39] examined the relationship between economic growth, non-renewable energy use, greenhouse gas emissions, and renewable energy use in European nations. They found that while an increased utilization of renewable energy results in a decrease in greenhouse gas emissions, heightened energy consumption leads to an increase in emissions. Moreover, according to Maneejuk et al. [40], economic development that surpasses a certain threshold point can lead to declining $CO_2$ emissions.

Additionally, there is a positive correlation between economic growth and $CO_2$ emissions in developed Middle Eastern and North African nations, while emerging nations exhibit a negative correlation [41]. Conversely, the findings of Anwar et al. [42] suggested that increasing economic expansion raises carbon dioxide emissions in G7 countries, subsequently exacerbating environmental pollution. However, they also found that higher levels of renewable energy consumption, institutional quality enhancement, and tech-

nological innovation hindered carbon dioxide emissions. Furthermore, Sanli et al. [43] demonstrated that in the long run, both macroeconomic stability and instability positively influence carbon emissions, with asymmetric economic complexity shocks exacerbating environmental pollution. Their study suggested that a sophisticated, complex production structure may have less detrimental effects on the environment compared to conventional production methods.

Nonetheless, Zhu et al. [44] suggested that an increase in government focus on environmental protection may moderate corporate efforts to reduce carbon emissions and enhance environmental subsidy provisions. This observation is echoed by Cao et al. [45], who highlighted that coordinated actions between local and central governments on carbon reduction are achieved through attention to environmental protection regulations. Furthermore, Liu et al. [46] found that businesses in capital-intensive, technologically advanced, and highly polluting industries demonstrate a more pronounced performance in carbon reduction. Additionally, attention to carbon reduction and financial market stress is noted to be closely associated with volatility spillover [47].

This study aims to fill existing research gaps by investigating the relationship between CCUS technology and carbon emissions in the US, while also exploring various predictors related to macroeconomics and energy fields. Its contributions are two-fold. Firstly, it enriches the literature on low-carbon transformation by offering practical insights for the US in developing low-carbon technology. While previous studies have highlighted the effectiveness of CCUS technology in reducing carbon emissions, few have focused specifically on the US context. This study addresses this gap by providing valuable insights tailored to the US. Secondly, it uncovers key factors that enhance CCUS capacity and reduce carbon emissions. By integrating various economic and energy factors into a comprehensive research framework and analyzing their impacts on both CCUS technology and carbon emissions, this study offers actionable solutions for policymakers to implement more effective measures toward carbon emission reduction and carbon neutrality.

## 3. Methodology

### 3.1. Estimation Methods

#### 3.1.1. Ordinary Least Squares (OLS)

The relationship between a dependent variable and one or more independent variables is analyzed through the multiple linear regression model [48,49]. The general form of the linear regression model is

$$y = X\beta + \varepsilon \tag{1}$$

where $y_t$ is the $T \times 1$ vector representing the dependent variable, and $X$ is $T \times K$, a matrix of independent variables. $\beta$ is the $K \times 1$ vector of unknown parameters, and $\varepsilon$ is the $T \times 1$ vector representing random disturbance that follows a normal distribution with a mean of 0 and variance $\sigma^2$, denoted as $\varepsilon \sim iidN(0, \sigma^2)$. OLS is the most commonly used method for estimating the unknown parameters in a linear regression model [48,49]. Then, the OLS estimator is $\hat{\beta}^{OLS} = \underset{\beta}{\mathrm{argmin}} \|y - x\beta\|$, where $\|.\|$ is the standard $L^2$ norm in the $T$-dimensional Euclidean space.

#### 3.1.2. Lasso Regression

The least absolute shrinkage and selection operator (Lasso) was developed by Tibshirani [50] for regression analysis where the number of predictors is larger than observations, i.e., $p > T$. Lasso performs not only the selection of variables but also parameter estimation to improve the prediction precision and interpretability of the generated statistical model.

Let $\hat{\beta}^{lasso} = (\hat{\beta}_1^{lasso}, \ldots, \hat{\beta}_k^{lasso})$, and the Lasso estimate $\hat{\beta}^{lasso}$ is defined as the solution to the following optimization problem:

$$\hat{\beta}^{lasso} = \underset{\beta^{lasso}}{\text{argmin}} \left\{ \sum_{t=1}^{T} \left( y_t - \sum_{k=1}^{K} \beta_k^{lasso} x_{kt} \right)^2 \right\}$$

$$\text{subject to } \sum_{k=1}^{K} \left| \beta_k^{lasso} \right| \leq c \qquad (2)$$

where $c \geq 0$ is the tuning parameter and for all $t$, the Lasso function can also be expressed in compacted form as

$$\hat{\beta}^{lasso} = \underset{\beta^{lasso}}{\text{argmin}} (y - X\beta^{lasso})^2 + \lambda \left\| \beta^{lasso} \right\|_1 \qquad (3)$$

where $\lambda$ is a non-negative regularization parameter and the exact relationship between $\lambda$ and the number of non-zero coefficients, denoted as $M$, is data-dependent.

### 3.1.3. Ridge Regression

Ridge regression, presented by Hoerl and Kennard [51] is a technique for calculating the coefficients of multiple-regression models in cases where predictors are highly correlated, i.e., multicollinearity issue. The Ridge regression is formulated as

$$\hat{\beta}^{ridge} = \underset{\beta^{ridge}}{\text{argmin}} \left\{ \sum_{t=1}^{T} \left( y_t - \sum_{k=1}^{K} \beta_k^{ridge} x_{kt} \right)^2 \right\}$$

$$\text{subject to } \sum_{j=1}^{p} \left( \beta_j^{ridge} \right)^2 \leq c \qquad (4)$$

The Ridge function can be stated alternatively as

$$\hat{\beta}^{ridge} = \underset{\beta^{ridge}}{\text{argmin}} (y - X\beta^{ridge})^2 + \lambda \left\| \beta^{ridge} \right\|_2^2 \qquad (5)$$

### 3.1.4. Elastic Net Regression

Elastic Net is a regularization regression method that combines penalties from both the Lasso and Ridge methods [52]. It is a suitable method for cases involving many predictors and multicollinearity. The Elastic Net regression can be defined as

$$\hat{\beta}^{elastic} = \underset{\beta^{elastic}}{\text{argmin}} \left\{ \sum_{t=1}^{T} \left( y_t - \sum_{k=1}^{K} \beta_k^{elastic} x_{kt} \right)^2 \right\}$$

$$\text{subject to } (1 - \alpha) \left\| \beta^{elastic} \right\|_1 + \alpha \left\| \beta^{elastic} \right\|_2^2 \leq c \qquad (6)$$

where $\left\| \beta^{elastic} \right\|_2^2 = \sum_{k=1}^{K} \left( \beta_k^{elastic} \right)^2$, $\left\| \beta^{elastic} \right\|_1 = \sum_{k=1}^{K} \left| \beta_k^{elastic} \right|$, and $\alpha$ is the weight parameter. The Elastic Net estimator $\hat{\beta}^{elastic}$ can be revised as

$$\hat{\beta}^{elastic} = \underset{\beta^{elastic}}{\text{argmin}} (y - X\beta^{elastic})^2 + \lambda_1 \left\| \beta^{elastic} \right\|_1 + \lambda_2 \left\| \beta^{elastic} \right\|_2^2 \qquad (7)$$

Elastic Net regression stands out as a robust regularization technique, offering solutions to several drawbacks encountered in Lasso and Ridge regression approaches. Its versatility makes it particularly valuable in datasets characterized by high dimensionality

and multicollinearity. Elastic Net effectively combines variable selection and shrinkage, making it a powerful tool for predictive modeling in complex data scenarios.

### 3.2. Data

This section outlines the variables utilized in this study, categorized into three groups. The first group encompasses CCUS and carbon emission-related variables, including CCUS capacity (CCUSCap), number of CCUS projects (Proj), CCUS technology level (Techlv), government CCUS policy (Govpol), and carbon emissions (CO2E). The second group comprises economic indicators, such as GDP per capita (Gdppc), unemployment rate (Unemp), trade volume (Trad), foreign direct investment (FDI), and industrial production and capacity utilization (Indprod). The third group consists of energy factors, including prices of coal (CoalP), oil (OilP), and natural gas (NgP), as well as energy consumption of fossil fuels (ECFF) and renewable energy (ECRE).

Table 2 presents the variables along with their respective data sources and literature references. The focus of this research is on the United States CCUS capacity and carbon emissions, along with their associated predictors, spanning from 1972 to 2022, encompassing yearly data and a total of 51 observations. The United States was chosen due to its extensive CCUS experience, with over 50 years of involvement and possession of nearly half of the world's CCUS capacity [3].

**Table 2.** List of variables, data sources, and related literature.

| Name | Description | Source | Authors |
|---|---|---|---|
| CCUSCap | Carbon Capture, Utilization, and Storage capacity (Mt $CO_2$/yr) | International Energy Agency | |
| Proj | Carbon Capture, Utilization, and Storage projects (active project/yr) | International Energy Agency | |
| Techlv | Carbon Capture, Utilization, and Storage technology level (new patent publication/yr) | European Patent Office | [14,53] |
| Govpol | Government policy on CCUS (active policy/yr) | Climate Policy Database | [54,55] |
| CO2E | Carbon dioxide emission from fuel combustion (Bt $CO_2$/yr) | Our World in Data | [14,56–60] |
| Gdppc | Gross domestic product (GDP) per capita (current USD) | World Bank | [14,56–60] |
| Unemp | Unemployment (%) | World Bank | [61,62] |
| Trad | Trade (% of GDP) | World Bank | [63–65] |
| FDI | Foreign direct investment (net inflows, balance of payment, millions current USD) | World Bank | [14,36,56,65,66] |
| Indprod | Industrial production and capacity utilization (index) | Federal Reserve Board | [67,68] |
| CoalP | Coal price (Australia market, in USD, real price) | World Bank | [36,69,70] |
| OilP | Crude oil price (average spot price of Brent, Dubai, and West Texas Intermediate, equally weighed, real price) | World Bank | [36,70] |
| NgP | Natural gas price (US Henry hub, real price) | World Bank | [70,71] |
| ECFF | Primary energy consumption from fossil fuels (quadrillion Btu) | U.S. Energy Information Administration | [36,71] |
| ECRE | Primary energy consumption from renewable energy (quadrillion Btu) | U.S. Energy Information Administration | [36,71] |

Table 3 provides detailed statistics for each parameter. To maintain consistency in units and mitigate the influence of long-term stochastic trends or unit roots, the time series data are transformed into growth rates. Unit root and multicollinearity tests are also conducted using the variance inflation factor (VIF).

This study examines the asymmetric distribution of variables by scrutinizing their skewness. From the data showcased in Table 3, it becomes evident that certain variables, including CO2E, Gdppc, Trad, FDI, Indprod, NgP, ECFF, and ECRE, exhibit negative skewness. This negative skewness suggests a downward trend in economic, energy, and carbon-related factors over time. Essentially, it indicates that the majority of observations have lower values, signifying a decline in these factors. Conversely, variables such as

CCUSCap, Proj, Techlv, Govpol, Unemp, CoalP, and OilP portray a positive trend in associated indicators. These variables suggest an upward trajectory in their respective factors. Furthermore, the outcomes of Jarque–Bera normality tests reveal that, apart from Techlv, Gdppc, FDI, and NgP, the data do not adhere to a normal distribution. Moreover, nearly all variables exhibit kurtosis values surpassing 3, indicating the presence of excess kurtosis. This implies that the distributions have heavier tails than a normal distribution, highlighting potential outliers or extreme values.

**Table 3.** Summary of descriptive statistics.

| Variable | Mean | Max | Min | SD | Skew | Kurt | Jarque–Bera Test | Unit Root Test | Equation (8) VIF | Equation (9) VIF |
|---|---|---|---|---|---|---|---|---|---|---|
| CCUSCap | 8.2057 | 137.7784 | −12.9397 | 25.4278 | 3.5757 | 16.2961 | 484.12 *** | −7.676 *** | NA | 3.80 |
| Proj | 6.7955 | 69.3147 | −7.4108 | 16.2677 | 2.6041 | 9.6403 | 152.2 *** | −8.514 *** | 1.33 | 3.52 |
| Techlv | 5.1427 | 105.4441 | −65.3927 | 34.5834 | 0.3044 | 3.2287 | 0.8683 | −8.767 *** | 1.24 | 1.20 |
| Govpol | 7.0265 | 109.8612 | −6.0625 | 19.8716 | 3.6620 | 17.1874 | 542.03 *** | −6.733 *** | 1.40 | 1.27 |
| CO2E | 0.2875 | 6.5732 | −11.0443 | 3.3981 | −0.8194 | 4.2425 | 8.786 ** | −7.037 *** | 72.84 | NA |
| Gdppc | 5.1189 | 11.1254 | −2.8720 | 3.0489 | −0.0675 | 3.2040 | 0.0341 | −4.842 *** | 3.24 | 2.54 |
| Unemp | −0.9416 | 78.5480 | −40.8576 | 19.8243 | 1.5619 | 7.2747 | 77.149 *** | −5.951 *** | 3.49 | 3.46 |
| Trad | 1.8307 | 22.8996 | −18.7570 | 7.0074 | −0.1415 | 4.4480 | 4.959 * | −6.688 *** | 7.92 | 9.10 |
| FDI | 12.0074 | 116.7361 | −96.3151 | 49.9526 | −0.1843 | 2.4578 | 0.8858 | −9.024*** | 2.11 | 2.14 |
| Indprod | 1.9482 | 9.1821 | −12.0660 | 4.3036 | −1.0988 | 4.5978 | 14.299 *** | −6.050 *** | 8.18 | 8.69 |
| CoalP | 7.2200 | 91.5619 | −57.0508 | 29.1879 | 0.8164 | 4.0419 | 5.3744 * | −5.345 *** | 2.67 | 2.58 |
| OilP | 7.9432 | 136.3325 | −63.9827 | 33.2699 | 0.9264 | 6.5674 | 38.968 *** | −6.409 *** | 4.08 | 4.95 |
| NgP | 6.9924 | 64.8129 | −80.7442 | 28.1008 | −0.3691 | 3.7934 | 3.2464 | −6.689 *** | 3.16 | 2.38 |
| ECFF | 0.3829 | 5.7313 | −9.4953 | 3.1671 | −0.7053 | 3.6409 | 4.799 * | −6.799 *** | 69.97 | 3.99 |
| ECRE | 2.4209 | 1.3869 | −15.3743 | 5.1674 | −0.7268 | 4.7471 | 10.15 *** | −6.806 *** | 1.36 | 1.26 |

Note: ***, **, and * in the Jarque–Bera and unit root tests mean *p*-value ≤ 0.01, ≤0.05, and ≤0.1. "NA" implies that the variable is a dependent variable.

Additionally, none of the variables exhibit a unit root, suggesting the absence of non-stationarity. This implies that the variables are stationary over time, without any significant long-term trend. Concerning multicollinearity, in CCUSCap Equation (8), high multicollinearity is solely observed between CO2E and ECFF. This indicates a strong relationship between these two variables, potentially necessitating caution in their interpretation. Conversely, in the case of CO2E Equation (9), no multicollinearity is identified, with all variance inflation factors (VIFs) registering as less than 10. This suggests that the variables in this equation are relatively independent of each other, reducing the risk of multicollinearity-related issues.

### 3.3. Empirical Model

In this study, several economic and energy factors are included to analyze the determinants of CCUS capacity and carbon emissions. Our empirical models are presented in Equations (8) and (9), respectively. To address potential endogeneity issues in our empirical study, we introduce lagged independent variables into our equations. Specifically, we include lagged terms for the independent variables to mitigate any endogeneity concerns. Equation (8) now integrates lagged independent variables alongside energy, economic, carbon emissions, and other related factors to elucidate the relationship with CCUS capacity. Similarly, Equation (9) incorporates lagged independent variables to explore the associations between carbon emissions and CCUS capacity, as well as energy, economic, and carbon-related variables.

$$\begin{aligned} CCUSCap_t = \beta_0 &+ \beta_1 Proj_{t-1} + \beta_2 Techlv_{t-1} + \beta_3 Govpol_{t-1} + \beta_4 CO2E_{t-1} + \beta_5 Gdppc_{t-1} \\ &+ \beta_6 Unemp_{t-1} + \beta_7 Trad_{t-1} + \beta_8 FDI_{t-1} + \beta_9 Indprod_{t-1} + \beta_{10} CoalP_{t-1} \\ &+ \beta_{11} OilP_{t-1} + \beta_{12} NgP_{t-1} + \beta_{13} ECFF_{t-1} + \beta_{14} ECRE_{t-1} + \varepsilon_{it} \end{aligned} \qquad (8)$$

$$\begin{aligned} CO2E_t = \beta_0 &+ \beta_1 Proj_{t-1} + \beta_2 Techlv_{t-1} + \beta_3 Govpol_{t-1} + \beta_4 CCUSCap_{t-1} + \beta_5 Gdppc_{t-1} \\ &+ \beta_6 Unemp_{t-1} + \beta_7 Trad_{t-1} + \beta_8 FDI_{t-1} + \beta_9 Indprod_{t-1} + \beta_{10} CoalP_{t-1} \\ &+ \beta_{11} OilP_{t-1} + \beta_{12} NgP_{t-1} + \beta_{13} ECFF_{t-1} + \beta_{14} ECRE_{t-1} + \varepsilon_{it} \end{aligned} \qquad (9)$$

## 4. Results and Discussion

### 4.1. Model Selection

In this research, we have employed four econometric estimations: Ordinary Least Squares (OLS), Lasso regression, Ridge regression, and Elastic Net models. The selection criterion for determining the most suitable model is based on identifying the one that exhibits the best performance. To assess this, we consider the Akaike Information Criterion (AIC) and Bayesian Information Criterion (BIC) values, with lower values indicating better performance.

Table 4 presents the results of the AIC and BIC values for each model. Remarkably, the Elastic Net model outperforms the others, demonstrating the lowest AIC and BIC values in the CCUSCap Equation (8). Conversely, OLS performs comparatively better than other models in the case of CO2E Equation (9). Therefore, the Elastic Net model and OLS are selected as the statistical models of choice for the CCUSCap and CO2E cases, respectively. This selection aligns with findings from previous studies indicating that the Elastic Net model performs better than OLS, Lasso, and Ridge regressions in the presence of multicollinearity, as observed in the CCUSCap case. On the other hand, the lack of multicollinearity in the CO2E case means OLS provides better performance. This is because regularization bias may not be necessary in the absence of multicollinearity, and OLS is capable of reducing both bias and variance, thereby exhibiting the smallest variance.

**Table 4.** Model selection.

| Model | CCUSCap | | CO2E | |
|---|---|---|---|---|
| | AIC | BIC | AIC | BIC |
| OLS | 1303.9109 | 1332.8882 | 832.0914 | 861.0687 |
| Lasso | 845.0513 | 852.7786 | 1050.6142 | 1066.0688 |
| Ridge | 973.6168 | 1002.5942 | 1013.4992 | 1042.4765 |
| Elastic Net | 794.1751 | 801.9024 | 1056.1488 | 1073.5352 |

### 4.2. The Impact of Carbon Emission and Related Factors on CCUS Technology

Based on the findings presented in Table 5, several noteworthy insights emerge regarding CCUSCap.

**Table 5.** Elastic Net regression results for CCUS capacity estimation.

| Variable | Elastic Net |
|---|---|
| Proj | 6.0560 |
| Techlv | - |
| Govpol | - |
| CO2E | - |
| Gdppc | - |
| Unemp | - |
| Trad | - |
| FDI | 1.5704 |
| Indprod | - |
| CoalP | - |
| OilP | −0.3398 |
| NgP | - |
| ECFF | - |
| ECRE | - |
| Constant | - |

Note: "-" denotes the variable is not selected by the model. Standard errors are shown in parentheses.

Firstly, positive impacts on CCUSCap are observed in the case of Proj and FDI. Specifically, a 1% increase in Proj is associated with a substantial 6.0560% increase in CCUSCap, highlighting the significant role of project expansion in enhancing the country's CCUS

capabilities for carbon reduction. This underscores the importance of investment in projects aimed at carbon capture and storage technologies as a means to address carbon emissions effectively. Additionally, FDI exhibits a positive effect, with a 1% increase in FDI leading to a noticeable 1.5704% increase in CCUSCap. This can be attributed to the decarbonization efforts in hard-to-abate sectors such as cement manufacturing, chemical production, and steelmaking, which lack alternative low-carbon technology options apart from CCUS. In response to this challenge, these sectors are driven to invest more in CCUS technologies, facilitated by the net inflows of FDI and the increasing number of CCUS projects. Notably, the US government's provision of performance-based tax credits for carbon capture projects further incentivizes such investments [72].

Conversely, a negative effect on CCUSCap is detected in OilP. A 1% rise in oil prices is associated with a 0.3398% reduction in CCUSCap. This finding contrasts with expectations, as higher oil prices typically stimulate investment in CCUS technologies, particularly due to their relevance in enhancing oil recovery processes. One plausible rationale is that despite the presence of higher oil prices, economic uncertainty or market volatility may be prompting companies to exercise caution and withhold long-term investments in technologies like CCUS. This hesitance could arise from concerns regarding future economic conditions or regulatory uncertainties, leading companies to prioritize short-term stability over long-term sustainability initiatives. Thus, while higher oil prices typically signal opportunities for CCUS investment, prevailing economic dynamics may be exerting a counteracting influence, contributing to the observed negative correlation between oil prices and CCUSCap. In terms of magnitude, among these three variables, it is evident that Proj plays a pivotal role in increasing CCUSCap. An intriguing finding pertains to the limited impact of Techlv. Surprisingly, the level of CCUS technology, as measured by the count of CCUS technology-related patents, does not demonstrate any significant influence on the CCUS capacity. This suggests that increasing technological advancements in CCUS may not necessarily lead to a corresponding increase in the CCUS capacity, despite Techlv promoting carbon emission efficiency.

Unexpectedly, Govpol did not drive the increase in the US's CCUS capacity over 50 years, as earlier policy instruments implicitly and partly covered CCUS until the enactment of the 45Q tax credit under the Energy Improvement and Extension Act in 2008 [9]. Although the 45Q tax credit underwent major reform in 2018, the US still lacks a legislature-based climate strategy at the federal level [8]. Additionally, we cannot statistically deduce the relationship between CO2E and CCUSCap, given the disparity between CO2E and CCUS expansion over five decades. During this time, CCUS was primarily used for enhanced oil and gas recovery instead of the carbon reduction option [73].

Within the economic indicators, only FDI shows a significant effect on the CCUS capacity, while Gdppc, Unemp, Trad, and Indprod do not exhibit any notable impact on the CCUS capacity. Similarly, among energy factors, only OilP demonstrates a discernible influence on the CCUS capability, while variables such as CoalP, NgP, ECFF, and ECRE do not show significant effects in our study on the CCUS capacity. This observation can be attributed to various factors such as a high failure rate, insufficient financial investment, limited market opportunities, and regulatory constraints surrounding CCUS projects, which hinder their widespread adoption [23–25]. Despite the United States' extensive history of injecting $CO_2$ into the subsurface for over half a century, its primary challenge lies more in policy and economic realms rather than technological constraints [9]. Therefore, the successful implementation of CCUS necessitates the establishment of robust legal and regulatory frameworks to ensure the effective management of CCUS operations and the safe storage of $CO_2$ [74]. Ultimately, achieving decarbonization goals will require ensuring that green alternatives are economically viable for businesses, encouraging their voluntary adoption of more affordable fossil fuels [75]. Nonetheless, CCUS remains a crucial technology for reducing emissions, and our findings underscore the importance of promoting CCUS projects to enhance the CCUS capacity. This observation aligns with

expert opinions advocating for the acceleration of CCUS alongside the development of renewable energy options [30].

### 4.3. The Impact of CCUS Technology and Related Factors on Carbon Emission

The estimation results are summarized in Table 6. Positive impacts on carbon emissions are observed for several factors, including Govpol, Gdppc, NgP, and ECFF. Specifically, a 1% increase in these factors leads to a growth in carbon emissions by 0.0063%, 0.0952%, 0.0131%, and 0.9999%, respectively. These findings underscore the significant influence of key factors driving the growth in carbon emissions in the US.

Notably, ECFF, Gdppc, and NgP emerge as pivotal contributors to the rise in carbon emissions. These results are consistent with prior research, which has also emphasized the positive effect of non-renewable energy consumption, energy prices, and GDP per capita on carbon emissions [76,77].

**Table 6.** Results of $CO_2$ emission estimation using OLS.

| Variable | OLS |
| --- | --- |
| CCUSCap | 0.0055 (0.0050) |
| Proj | 0.0011 (0.0075) |
| Techlv | 0.0018 (0.0021) |
| Govpol | 0.0063 * (0.0037) |
| Gdppc | 0.0952 ** (0.0341) |
| Unemp | −0.0012 (0.0061) |
| Trad | 0.0221 (0.0281) |
| FDI | −0.0021 (0.0019) |
| Indprod | −0.0255 (0.0447) |
| CoalP | −0. 0032 (0.0036) |
| OilP | −0.0026 (0.0044) |
| NgP | 0.0131 *** (0.0036) |
| ECFF | 0.9999 *** (0.0412) |
| ECRE | −0.0265 * (0.0142) |
| Constant | −0.6058 *** (0.1716) |

Note: Significance at 0.01, 0.05, and 0.1 level indicated by ***, **, and *, respectively. Standard errors are shown in parentheses.

Remarkably, Govpol shows a positive correlation with CO2E, indicating that government policy on CCUS does not effectively reduce carbon emissions. This finding reinforces the notion that policy and economic factors, rather than technological ones, are the primary barriers to emission reduction [9]. On the contrary, a negative impact on carbon emissions is observed for ECRE. A 1% rise in ECRE leads to a reduction in carbon emissions by 0.0265%. These results align with previous research that has investigated the impact of ECFF and ECRE on greenhouse gas emissions [36,37]. The findings underscore that increasing the use of renewable energy sources would contribute to a decrease in $CO_2$ emissions.

Due to the higher *p*-values of CCUSCap, Proj, Techlv, Unemp, Trad, FDI, Indprod, CoalP, and OilP, they are not assumed to have significant impacts on carbon emissions, as indicated by our findings in Table 6. It is interesting to note that, despite being a carbon reduction technology, CCUSCap should theoretically have a negative relationship with CO2E. However, this expected relationship has not been observed [26]. This suggests that CCUSCap alone may not be sufficient to effectively reduce $CO_2$ emissions, possibly due to its relatively small contribution—less than 0.5% of the overall US $CO_2$ emissions, as depicted in Figure 3. Furthermore, Techlv and Proj also exhibit non-significant relationships with CO2E. When considering all the CCUS indicators—CCUSCap, Proj, Techlv, and Govpol—to CO2E, the overall effectiveness of CCUS in carbon reduction is questionable. This highlights the existing barriers to CCUS adoption in the US.

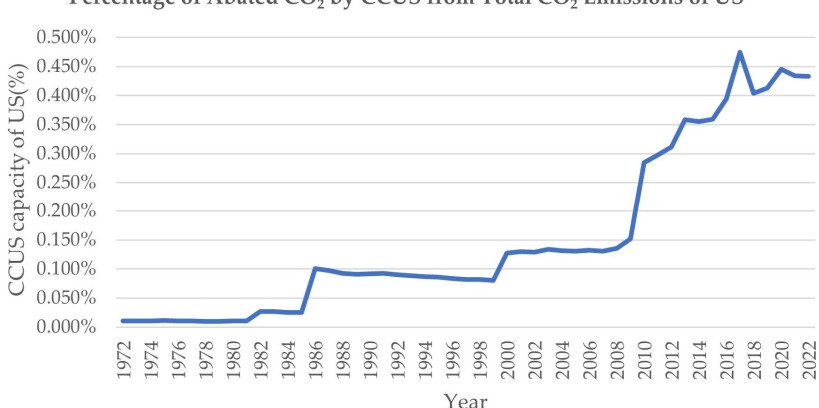

**Figure 3.** The percentage of reduced $CO_2$ by CCUS from the US's total carbon emission [3].

Overall, we observed that renewable energy can reduce carbon emissions while CCUS is not well-developed enough to contain carbon emissions as its capacity is very limited. However, CCUS has the potential to abate carbon emissions when it is widely applied, and its costs are lower than in the case of renewable energy [77].

*4.4. Robustness Checks*

To ensure the robustness of our results from the previous section, we re-estimated our models using the Generalized Method of Moments (GMM) estimation. The GMM is a flexible estimation method commonly employed in econometrics, especially when the underlying distribution of the data is unknown or challenging to specify. In the context of our models, the GMM proves particularly valuable in addressing issues such as endogeneity, serial correlation, or heteroscedasticity. Furthermore, we recognize that in estimating Equations (8) and (9) separately, there may be potential interdependencies between the equations that are overlooked. To address this concern and enhance the robustness of our results, we employ GMM estimation along with Seemingly Unrelated Regression (SUR) estimation. This approach allows us to simultaneously estimate both equations, thus capturing any potential interdependencies between them and providing more reliable results, as in Table 7.

**Table 7.** Robustness analysis.

|  | SUR | | GMM | |
| --- | --- | --- | --- | --- |
|  | **CCUSCap** | **CO2E** | **CCUSCap** | **COE2** |
| CCUSCap |  | 0.0055 (0.0048) |  | 0.0054 * (0.0029) |
| CO2E | 5.8352 (5.3742) |  | 5.1253 (3.8895) |  |
| Proj | 1.1689 *** (0.1515) | 0.0011 (0.0072) | 1.1939 *** (0.2237) | 0.0014 (0.0056) |
| Techlv | 0.0064 (0.0689) | 0.0018 (0.0019) | 0.0065 (0.0668) | 0.0022 (0.0014) |
| Govpol | −0.0129 (0.1272) | 0.0063 * (0.0035) | −0.0131 (0.0589) | 0.0069 ** (0.0031) |
| Gdppc | −0.1166 (1.2637) | 0.0952 ** (0.0314) | −0.1192 (1.0950) | 0.1077 *** (0.0262) |
| Unemp | −0.1013 (0.2017) | −0.0012 (0.0062) | −0.1014 (0.1824) | 0.0044 (0.0069) |
| Trad | 2.1202 * (0.8593) | 0.0221 (0.0269) | 2.1392 ** (1.0700) | 0.022 (0.0172) |
| FDI | 0.0565 (0.0623) | −0.0021 (0.0019) | 0.0521 (0.0391) | −0.0010 (0.0019) |
| Indprod | −2.3183 (1.4221) | −0.0255 (0.042) | −2.3293 (1.5012) | −0.0281 (0.0391) |
| CoalP | −0.0882 (0.1198) | −0.0023 (0.0036) | −0.0792 (0.0846) | −0.0049 (0.0030) |
| OilP | −0.3701 ** (0.1299) | −0.0032 (0.0041) | −0.3934 * (0.1939) | −0.0026 (0.0025) |
| NgP | 0.0201 (0.1353) | 0.0131 *** (0.0034) | 0.0204 (0.1057) | 0.0123 *** (0.0025) |
| ECFF | −5.6612 (5.6514) | 0.9999 *** (0.0404) | −5.6384 (4.1727) | 1.0195 *** (0.0377) |
| ECRE | −0.4462 (0.4822) | −0.0265 * (0.0136) | −0.4463 (0.3892) | −0.0270 ** (0.0115) |
| Constant | 5.7847 (6.7422) | −0.6058 *** (0.1543) | 5.7840 (6.0982) | −0.7034 *** (0.1453) |

Note: Significance at 0.01, 0.05, and 0.1 level indicated by ***, **, and *, respectively. Standard errors are shown in parentheses.

Based on the results obtained from the GMM and SUR estimations, it is evident that the estimated coefficients and their significance align with the findings of our main estimations presented in Tables 5 and 6. This consistency underscores the robustness of our results from the previous section.

Additionally, as we have not found evidence of a relationship between CCUS technology and carbon emissions, we further validate this result by employing the Granger causality test [78]. The results of this test are reported in Table 8. Notably, the findings indicate that we do not observe a significant impact of the CCUS capacity on $CO_2$ emissions, and vice versa. This reaffirms the results obtained from the main model, providing additional support for our conclusions.

**Table 8.** Granger causality test.

| Hypothesis | Wald Statistic |
|---|---|
| CO2E does not cause CCUSCap | 2.5208 |
| CCUSCap does not cause CO2E | 1.7738 |

## 5. Conclusions

This study aims to explore the relationship between the Carbon Capture, Utilization, and Storage (CCUS) capacity and carbon emissions in the US from 1972 to 2022. Employing four distinct statistical methods and considering thirteen relevant predictors, we endeavor to deepen our empirical understanding of this complex relationship. To enhance the scope of our investigation, we introduce two novel variables, CCUSCap and Proj, which provide additional insights into the dynamics at play. Notably, our analysis reveals that the Elastic Net model proves most effective in elucidating the CCUS capacity, whereas the OLS model is better suited for understanding carbon emissions. Our findings are robust and further validated through GMM and SUR estimations, providing confidence in the reliability of our results. Through this study, we contribute to the existing body of knowledge in the field of CCUS and carbon emissions, shedding light on the key factors driving or hindering progress in these critical areas of environmental concern.

The findings regarding the CCUS capacity equation suggest that while OilP has a negative impact on the US's CCUS capacity, Proj and FDI show significant positive correlations. Specifically, Proj emerges as one of the most influential factors among the variables analyzed, demonstrating a substantial positive effect on raising CCUSCap. Interestingly, the analysis also reveals that technological advancements in CCUS may not always translate into an increased CCUS capacity. The 50-year increase in the US CCUS capacity was unexpectedly not driven by Govpol. Moreover, the widening gap between CO2E and CCUS over five decades, coupled with CCUS's predominant use in enhanced oil and gas recovery rather than carbon reduction, makes it difficult to statistically determine the relationship between CO2E and CCUSCap. Nevertheless, our results underscore the importance of prioritizing Proj to enhance CCUSCap. This aligns with the perspectives of CCUS experts who emphasize the necessity of accelerating both renewable energy sources and CCUS technologies.

In the case of the carbon emissions equation, our analysis reveals that an increase in ECRE leads to a reduction in carbon emissions in the US. Conversely, Govpol, Gdppc, NgP, and ECFF are associated with increases in carbon emissions. CCUSCap is theoretically expected to have a negative correlation with CO2E as a carbon reduction technology; our study does not observe this relationship. This suggests that CCUSCap alone may not be sufficient to effectively reduce $CO_2$ emissions. This discrepancy could be attributed to the significant disparity between the amount of $CO_2$ captured and the total $CO_2$ emissions, where the captured $CO_2$ accounts for less than 0.5% of the total US $CO_2$ emissions. Considering all the CCUS indicators—CCUSCap, Proj, Techlv, and Govpol—in conjunction with CO2E, the overall effectiveness of CCUS in reducing carbon emissions appears uncertain.

However, if CCUS becomes more widely adopted and cost-effective compared to renewable energy sources, it could potentially play a significant role in reducing carbon emissions.

Despite more than 50 years of injecting $CO_2$ underground in the US, the real challenge lies in policy and economic considerations rather than technological limitations. To ensure the effective management of CCUS operations and the safe storage of $CO_2$, it is imperative to develop legal and regulatory frameworks. Decarbonization efforts will be most successful when green alternatives become economically viable for businesses, encouraging the voluntary adoption of fossil fuels. It is important to recognize that CCUS is not a standalone solution but rather a complementary approach to carbon emission reduction. To achieve net zero goals, a combination of strategies, including renewable energy adoption, energy efficiency improvements, and behavioral changes, will be necessary.

Our study has several limitations that warrant consideration. Firstly, we solely employed the linear regression model, neglecting the potential impacts of thresholds. Future research could explore threshold regression techniques to better capture nonlinear relationships in the data. Secondly, our focus was limited to the United States, which may restrict the generalizability of our findings to other countries. Extending the analysis to include countries with advanced CCUS technology could provide a more comprehensive understanding of the factors influencing the CCUS capacity and carbon emissions on a global scale.

**Author Contributions:** Conceptualization, M.T.M. and R.T.; methodology, M.T.M., R.T., and W.Y.; software, M.T.M. and W.Y.; validation, M.T.M., R.T., and W.Y.; formal analysis, M.T.M.; investigation, M.T.M.; resources, M.T.M.; data curation, M.T.M.; writing—original draft preparation, M.T.M. and R.T.; writing—review and editing, M.T.M., R.T., and W.Y.; visualization, M.T.M. and R.T.; supervision, R.T. and W.Y.; funding acquisition, R.T. All authors have read and agreed to the published version of the manuscript.

**Funding:** This research received no external funding.

**Data Availability Statement:** The original contributions presented in the study are included in the article, further inquiries can be directed to the corresponding author.

**Acknowledgments:** The first author would like to acknowledge the master's degree program in Economics in the Faculty of Economics, Chiang Mai University, under the CMU Presidential Scholarship. This research work was partially supported by Chiang Mai University, Thailand.

**Conflicts of Interest:** The authors declare no conflicts of interest.

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
