# Peer review of "CCUS Technology and Carbon Emissions: Evidence from the United States"

_energies, doi:10.3390/en17071748_

Round 1

Reviewer 1 Report

Comments and Suggestions for Authors

I suggest to shorten the Introduction; the first paragraphs have the sole scope of motivating the need of CCUS and can be resumed in a single (or two maximum) sentence.

Similarly, also the abstract should be slightly compacted.

In the caption of Figure 2, the ref number can be directly inserted at the end of the sentence, without using "Source:".

In Section 2, the authors should insert a table showing the main CCUS techniques and the quantity of CO2 annually stored with each of them.

The format of Section 3 needs to be revised.

Also the concluding section should be shortened.

Comments on the Quality of English Language

The quality of english is acceptable.

Reviewer 2 Report

Comments and Suggestions for Authors

This research explore the relationship between CCUS capability and carbon emission in the United States. The conclusion is that positive impacts of projects and foreign direct investment on CCUS capacity, while the influence of CCUS technology level is found to be limited. However, the relationship between CCUS capacity and carbon emissions remains limited. The study highlights the importance of incentivizing projects to increase CCUS capabilities and recognizes the critical role of legal and regulatory frameworks in facilitating effective CCUS implementation in the US. Moreover, the paper emphasizes that achieving decarbonization goals necessitates the development of affordable green alternatives. Overall, the motivation for the paper is good as it focuses on an important question. The story is interesting.

Here are some recommendations for improvement:

1. The introduction requires restructuring to better contextualize the major contributions of this research. The positioning of key findings and contributions should be clarified.

2. The literature review needs to be improved, some recent articles have investigated the carbon reduction attention effect should be included.

3. The Conclusions is too long. It should be refined in conjunction with this journal.

Round 2

Reviewer 1 Report

Comments and Suggestions for Authors

The article can now be considered for publication.

Comments on the Quality of English Language

English language is suitable for publication.